# Development of a constant pressure perfused *ex vivo* model of the equine larynx

**Sven Otto**[1☯¤]**, Jule K. Michler**[1☯]***, Stefan Dhein**[2]**, Christoph K. W. Mülling**[1]

**1** Institute of Veterinary Anatomy, Histology and Embryology, Faculty of Veterinary Medicine, Leipzig University, Germany, **2** Fachdienst Gesundheit, Altenburg, Germany

☯ These authors contributed equally to this work.
¤ Current address: Small Animal Clinic Panitzsch, Borsdorf, Germany
* jule.michler@vetmed.uni-leipzig.de

**Data Availability Statement:** All relevant data are within the paper and its Supporting Information files.

**Funding:** SO received a stipend through Med-El Company, Austria. Project-No: 2511-0712 (https://

## Abstract

Distal axonopathy is seen in a broad range of species including equine patients. In horses, this degenerative disorder of the recurrent laryngeal nerve is described as recurrent laryngeal neuropathy (RLN). The dysfunctional innervation of the cricoarytenoideus dorsalis muscle (CAD) leads to a loss of performance in affected horses. In general, *ex vivo* models of the larynx are rare and for equine patients, just one short report is available. To allow for testing new therapy approaches in an isolated organ model, we examined equine larynges in a constant pressure perfused setup. In order to check the vitality and functionality of the isolated larynx, the vessels´ reaction to norepinephrine (NE) and sodium nitroprusside (NP) as vasoactive agents was tested. Additionally, the contractility of the CAD was checked via electrical stimulation. To determine the extent of hypoxic alterations, lactate dehydrogenase (LDH) and lactate were measured and an immunofluorescent analysis of hypoxia-inducible factor (HIF-1α), a key transcription factor in hypoxia, was performed. For this, a hypoxia-induced cell culture for HIF-1α was developed. The application of NE led to an expected vasoconstriction while NP caused the expected vasodilation. During a perfusion period of 352 ±20.78 min, LDH values were in the reference range and lactate values slightly exceeded the reference range at the end of the perfusion. HIF-1α nuclear translocation could reliably be detected in the hypoxia-induced cell cultures, but not in sections of the perfused CAD. With the approach presented here, a solid basis for perfusing equine larynges was established and may serve as a tool for further investigations of equine larynx disorders as well as a transferrable model for other species.

## Introduction

Recurrent laryngeal neuropathy (RLN) in horses is a distal axonopathy of the recurrent laryngeal nerve which branches from the vagus nerve, the X. cranial nerve [1]. The distal axonopathy inevitably results in an insufficient motor innervation of the cricoarytenoideus dorsalis muscle (CAD), the only dilator of the vocal fold. Therefore, affected horses show a reduced or absent vocal fold abduction. This provokes a breathing noise, the characteristic roaring. The

www.medel.com/de). The specific roles of these authors are articulated in the 'author contributions' section.The authors also received financial support from Leipzig University for Open Access Publishing. The funders had no role in study design, data collection and analysis, decision to publish, or preparation of the manuscript.

**Competing interests:** The authors have read the journal's policy and the authors of this manuscript have the following competing interests: SO received a stipend through Med-El Company, Austria. There are no patents, products in development or marketing products to declare. This does not alter our adherence to PLOS ONE policies on sharing data and materials.The pilot study of this project was presented at the BMT (Biomedizinische Technik) congress "Dreiländertagung der Deutschen, Schweizerischen und Österreichischen Gesellschaft für Biomedizinische Technik, Graz, Austria" and published as a case study in the congress proceedings [42].

disorder occurs predominantly on the left side and rarely bilateral, causing neurogenic atrophy in the affected muscle tissue [2]. The prevalence for RLN in horses differs widely as summarized e.g. by Ducharme et al. [3]. Several surgical techniques have been established to symptomatically treat the affected vocal fold [4]. Currently, besides nerve transplantation [5], functional electrical stimulation (FES) is a promising strategy for the causal treatment of this disorder [3, 6–8]. Measuring the 3D distribution of the electrical field around implanted stimulation electrodes might be a useful method to optimize FES, as shown in a constant flow *ex vivo* model [9].

For ethical reasons, e*x vivo* organ models are an auspicious approach in order to meet the principles of the 3R concept. Moreover, they allow studies on organ level under standardized conditions and can be viewed as advantageously filling the gap between *in vitro* techniques (cell cultures or tissue explants) and *in vivo* experiments.

One of the first reports of a perfused *ex vivo* model was the Langendorff heart [10] from which numerous current techniques evolved that are in use nowadays. For the larynx, perfused *ex vivo* models of humans with a pulsatile flow approach [11], two constant flow models of dogs [12] and the above-mentioned study by Martini et al. [9] for horses are described.

Detached from the blood circulation, isolated organs are exposed to hypoxia. Testing for the organs´ viability and evaluation of hypoxic tissue damage is therefore crucial. As summarized [13], for muscle tissue with a reported ischemic tolerance from 4-6h, the muscles´ function itself (e.g. contractility) is a hallmark but should be supported by additional testings. Lactate is a well-known standard marker for tissue hypoxia [14, 15]. Upon cell membrane damage, lactate dehydrogenase (LDH) is released [16] and can be used for detecting cellular injury [17, 18].

As reviewed by Weidemann et al. [19] among others, hypoxia-inducible factor-1 alpha (HIF-1α) plays a central role regarding cell metabolism and the required cellular adaptations under hypoxic conditions in mammalian cell types. HIF-1α is studied in altitude research, ischemic reperfusion injury (IRI) and in a variety of cell culture assays, e.g. for psoriasis [20].

Under normoxic conditions, the α-subunit of HIF is hydroxylated and instantaneously degraded. This changes in hypoxia; the HIF-1α subunit is stabilized because its hydroxylation is suppressed and the protein shuttles to the nucleus to bind to HIF-1β [21].

This nuclear translocation can be induced in cell cultures using hypoxia incubators, deferoxamine or cobalt chloride [22–24] providing a reliable positive control for analyses regarding HIF-1α.

The work presented here aimed at developing a reliable perfused *ex vivo* model of the equine larynx through establishing a constant pressure perfusion with special regard to the CAD. The organ's vitality was monitored by electrical stimulation and by measuring the well-established markers LDH and lactate. Additionally, the micromorphological evaluation was substantiated by immunofluorescent analysis of HIF-1α in order to assess possible hypoxic alterations in the CAD of the perfused larynx.

## Materials and methods

Larynges from five adult mares (weight range 400–550 kg, age between 4 to 14 years) were used for this study. All horses were sound as confirmed by clinical examination and had no known history of RLN. Animals were euthanized in consent with the national guidelines for animal welfare. One horse was euthanized because of a chronic orthopedic disorder, the other four larynges were dissected from experimental animals. All procedures were carried out in accordance with the national animal welfare guidelines and were approved by the

Landesdirektion Sachsen (animal experiment number / ethic committee approval: TVV 34/13, W 09/14). Euthanasia was carried out by anaesthesists of the equine clinic (Leipzig University).

After intrajugular catheterization, detomidine (0.06 mg/kg) and butorphanol (0.03 mg/kg) were administered for sedation. Deep anaesthesia was achieved using ketamine (2.2 mg/kg) and diazepam (0.08 mg/kg). Finally, for euthanasia T61 (0.12 mL/kg) and heparin (100 IU/kg) were injected, the latter inhibiting blood clotting.

The larynx was dissected and isolated immediately post mortem. Particular attention was paid to spare the branching of the cranial thyroid artery from the common carotid artery. Fig 1 shows a schematic drawing of the equine laryngeal vascularization and the catheter placing. Fullest attention was paid to the supplying arteries during the dissection. It was taken care of that the vessel stumps were long enough to enable for an easy identification and catheterization. The larynx was then transferred to the laboratory in a plastic box containing a modified Tyrode solution (the perfusion solution, here at room temperature, RT) within 30 minutes. The saline Tyrode solution contained (in mmol/L): NaCl 137.1, KCl 4.02, $NaH_2PO_4$ 0.79, $MgCl_2$ 1.0, D-glucose 5.53, $NaHCO_3$ 24.04, $CaCl_2$ 2.66, sodium pyruvate 2.0. Furthermore, 1 IU/L insulin was added. The organ was connected to the perfusion system (Fig 1) and attached to a larynx retainer specifically designed for stable positioning of the organ. Subsequently, it was placed in a perfusion tub filled with perfusion solution (37˚C). Initially, an adaptation phase of 10 minutes was allowed for perfusing the larynx with a constant pressure of 7.85 kPa. After adaptation, the perfusion pressure was increased to 9.81 kPa. The perfusion solution was permanently gassed with Carbogen (95% $O_2$, 5% $CO_2$, Linde AG, Pullach, Germany). The perfusion flow was measured by a Doppler ultrasonography flowmeter (SONOFLOW CO55/080, SONOTEC Ultraschallsensorik Halle GmbH, Halle, Germany).

## Vitality tests

At the beginning and at the end of the experiment, the contractility of the CAD was examined. An insertion tool [25] was used for external electrical stimulation (1 Hz, 5 mA, 10 ms duration pulses, biphasic, 8 s). The counter electrode consisted of a small wire frame immersed in the perfusion tub. A pressure transducer (Transducer Model SPR-524, Millar Instruments, Inc., Houston, Texas, USA) was inserted into the medial part of the CAD.

The electrical impulses and the change of the intramuscular pressure during contraction were recorded by a PowerLab (PowerLab 4/25, ADInstruments, Dunedin, New Zealand) and a dedicated software program (LabChart 8, ADInstruments).

Vitality of the arterial vessels was checked by administration of vasoactive agents at the beginning and at the end of each experiment. Initially, the baseline flow (BL) was measured over a period of five minutes. The following five minutes were used to record the flow after the injection of norepinephrine (concentration of 0.2 mg/mL, 10mL total, Sigma Aldrich GmbH, Steinheim, Germany) as a vasoconstricting agent and the flow was recorded for another five minutes (NE). Then, sodium nitroprusside (with a concentration of 0.3 mg/mL, 10mL total, Sigma-Aldrich) were administered for vasodilatation and the perfusion flow was measured for five minutes (NP). Afterwards, the entire perfusion solution was substituted by fresh, warmed perfusion solution.

The concentration of lactate and LDH in the perfusion solution in the perfusion tub were measured at the beginning of the perfusion, before substitution of the perfusion solution and at the end of the experiment. Samples of 1–4 mL of the perfusion solution were collected in heparinized plastic syringes and stored at -20˚C for later analysis.

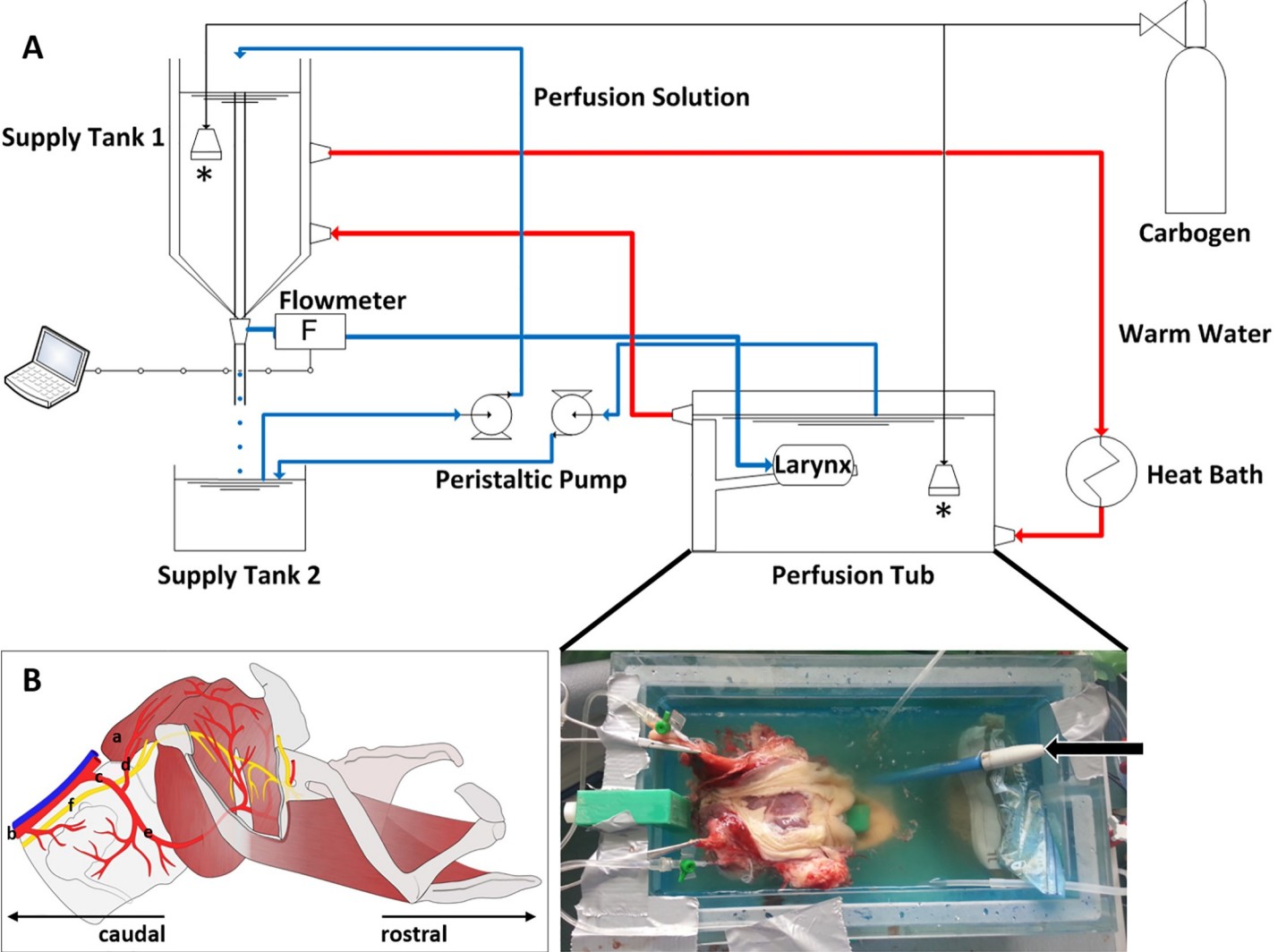

**Fig 1. Experimental setup.** A: The perfusion system (technical drawing) consists of a closed circuit. Red color indicates the hot water circuit, blue represents the perfusion circuit, gassed with carbogen. For active perfusion, the arteries were catheterized, passive venous outflow drains into the perfusion tub. The gassing tubes (one in the supply tank 1 and one in the perfusion tub) are denoted with an asterisk. The perfusion flow was measured using a flowmeter (F). In the inset photograph, a dorsal view of the perfusion setup *in situ* can be observed. The larynx, immersed in perfusion solution, is placed on a larynx retainer to stabilize its position in the perfusion tub. Both cranial thyroid arteries were catheterized (intravenous catheter, 18 gauge). For pH-control, a pH-meter is added (marked with an arrow). B: Schematic drawing of an equine larynx, lateral aspect (redrawn after our specimens and corrosions casts of the Institute of Veterinary Anatomy, Leipzig University). Relevant structures are the cricoarytenoideus dorsalis muscle [CAD, a)], the common carotid artery [b] running parallel to the external jugular vein], the catheterized arterial branch [c], cranial thyroid artery] followed by its ramification into the ascending pharyngeal artery (d) and the laryngeal branch (e). The latter two vessels nourish the CAD. f) Recurrent laryngeal nerve reaches the larynx as caudal laryngeal nerve.

For histological evaluation, muscle samples of approximately 1 cm x 1 cm were dissected from the left and right CAD after the perfusion. The obtained slides were compared to archived samples of unperfused equine CAD of the Institute of Veterinary Anatomy.

### Histological analysis, cell culture and hypoxia model

Muscle biopsies obtained after perfusion served for assessment of the general tissue morphology and immunohistochemical analyses. Briefly, specimens were fixed with 4% PFA, paraffin-embedded, cut using a rotation microtome and mounted on previously silanized SuperFrost

slides (VWR Int. GmbH, Dresden, Germany, cat.no. 631–0652). Tissue sections were deparaffinized and subsequently stained with Haematoxylin and Eosin (H&E) following a standardized procedure.

For indirect immunofluorescent analysis, heat-induced epitope retrieval (HIER) was performed as recommended in the data sheet of the anti-HIF-1α-antibody (rabbit polyclonal anti-HIF-1α, Abcam plc, Cambridge, UK, cat.no. ab114977) using citrate buffer (pH 6) in a steam cooker for 20 min. Tissue sections were blocked and permeabilized with 5% goat normal serum (GNS) and 0.2% Triton™X-100 in PBS for 45 min (GNS was obtained from Dianova GmbH, Hamburg, Germany, Triton™X-100 from AppliChem, Darmstadt, Germany).

The primary anti-HIF-1α-antibody was then applied (dilution 1:100 in PBS) and incubated for 4 h at RT. After washing with PBS three times, the secondary antibody (Alexa Fluor® 568 goat anti-rabbit, Dianova, cat.no. 711-545-152, 1:500 in PBS) was applied and incubated for 1 h at RT. The counterstaining of nuclei was carried out using Hoechst 33342 (BisBenzimide H 33342, Sigma Aldrich, cat.no. 14533). From a stock solution (1 mg/mL), a 1:1000 dilution was prepared in PBS and administered for 10 min at RT.

A control for unspecific binding of the secondary antibody control was stained simultaneously. Here, the primary antibody was omitted. For a thorough positive control, a cell culture hypoxia model was developed.

The aforementioned effects seen under hypoxic conditions can be mimicked chemically using cobalt(II) chloride ($CoCl_2$, cobalt(II) chloride hexahydrate, Sigma Aldrich, cat.no. C8661).

To assure that the chosen anti-HIF-1α-antibody showed cross reactivity in equine tissues, an induced hypoxia model in equine fibroblast cell culture modified from the protocol of Wu et al. [26] was established.

The fibroblasts were harvested from skin biopsies. These skin samples were declared waste material from a local abattoir (Freiberg, Germany). The cells had been previously isolated for experiments not related to this study and were kept in stock in liquid nitrogen. Equine fibroblasts from passages (P) 5–7 were seeded into 48 well plates in a DMEM/F12 medium (Thermo Fisher Scientific, Darmstadt, Germany) substituted with 10% fetal horse serum (FHS, Biowest, cat.no. S0960-500, distributed by Th. Geyer GmbH & Co KG, Renningen, Germany) and penicillin/streptomycin (Thermo Fisher Scientific). To determine suitable concentrations of $CoCl_2$ for induction of hypoxia in equine cells, preliminary tests were performed. Working dilutions were freshly prepared in cell culture medium immediately before the incubation. They were diluted from a 10 mM stock solution of $CoCl_2$ in ultrapure water. Concisely, concentrations from 100 μM/mL to 500 μM/mL were tested and incubation times varied (24, 36, 48 h). These initial experiments of three different primary isolations showed steadily induced hypoxia after 24 h using a 250 μM $CoCl_2$ solution in cell culture medium. Longer times led to loss of cells through apoptosis. For each test, a plate with untreated cells served as a negative control. The hypoxia induction was verified by immunocytochemical analysis for HIF-1α. For immunocytochemistry, the anti-HIF-1α-antibody was applied as described above after fixation (10 min PFA at RT) followed by 30 min blocking and permeabilization (this step consisted of treatment with 5% donkey normal serum (DNS) and 0.2% Triton™X-100 in PBS). The labeling secondary antibody used was an Alexa Fluor® 488 donkey anti-rabbit with a dilution of 1:500 in PBS. After repeated washing steps, the positive reaction was shortly checked under the microscope and the cells were then stained for intermediate filaments using a Vimentin-Cy™3-antibody (mouse monoclonal, Sigma Aldrich, cat.no. C9080, dilution 1:500 in PBS, 1h at RT).

Pictures were taken with a Nikon TE2000S inverse fluorescent microscope (Nikon GmbH, Düsseldorf, Germany) with motorized stage (Prior ProScan III) and imaging software NIS

elements AR (version 4.20, Nikon). The images were processed with the license free software imageJ and NIS elements viewer (version 4.20, Nikon).

For statistical analysis Microsoft® Excel 2010 (Microsoft Corporation, Redmond, USA) and SigmaPlot 11 (Systat Software GmbH, Erkrath, Germany) were used. Statistical evaluation was carried out using One-Way repeated measures ANOVA and the Holm-Sidak Test for post hoc analysis in case of significant differences. The significance level was set to $p < 0.05$ throughout.

## Results

The mean duration of the perfusion was approximately six hours (352 ±20.78 min).

The macroscopic examination after the perfusion revealed mild to moderate tissue edema. This was apparent in the vocal folds and the glottis.

### Electrical stimulation of the CAD

In all five horses, an increase of the intramuscular pressure in the left and right CAD could be triggered by external electrical stimulation at the beginning and at the end of the perfusion. The abduction of the vocal fold triggered by a contraction of the CAD was noted visually. Fig 2A shows a typical graph during the electrical stimulation and the corresponding muscular response.

### Vasoactive drug tests

Four larynges showed a decrease in the perfusion flow after the administration of NE mirroring vasoconstriction at the beginning of the experiment while at the end of the perfusion, all five larynges showed a vasoconstriction. NP caused an increase in the perfusion flow in four larynges at the beginning and in all five larynges at the end of the experiment reflecting vasodilation.

The application of NE and NP yielded no significant change in the perfusion flow at the initiation of the experiment. In contrast to that, the perfusion flow changed significantly at the end of the experiment when BL was compared to NE and NE compared to NP.

The mean values and the standard deviation of the perfusion flow are shown in Fig 2B (at the beginning of the perfusion) and Fig 2C (at the end of the perfusion).

### Lactate and LDH measuring

The concentration of LDH (mean ± SD) increased during the experiment (Fig 2D). The increases from LDH1 (beginning of the perfusion, 11.0 ± 8.49 U/L) to LDH3 (end of perfusion, 259.2 ± 172.76 U/L) and from LDH2 (before changing the perfusion solution, 31.0 ± 32.04 U/L) to LDH3 were both statistically significant.

Concentration of lactate (mean ± SD) in the perfusion solution of the perfusion tub showed an increase during the experiment (Fig 2E). There was a significant increase at the beginning of the perfusion (L1, 0.20 ± 0.03 mmol/L) compared to the time point before the perfusion solution was changed (L2, 0.58 ± 0.11 mmol/L) and compared to the end of the perfusion (L3, 1.68 ± 0.33 mmol/L). The increase of the lactate concentration was also significant when L2 was aligned to L3.

### Histology and immunofluorescent staining

According to the macroscopical findings, the tissue morphology of the CAD post perfusion in the H&E staining showed mild to moderate interstitial edema causing separated muscle fibers (Fig 3A [non-perfused CAD for comparison] and B [after perfusion]).

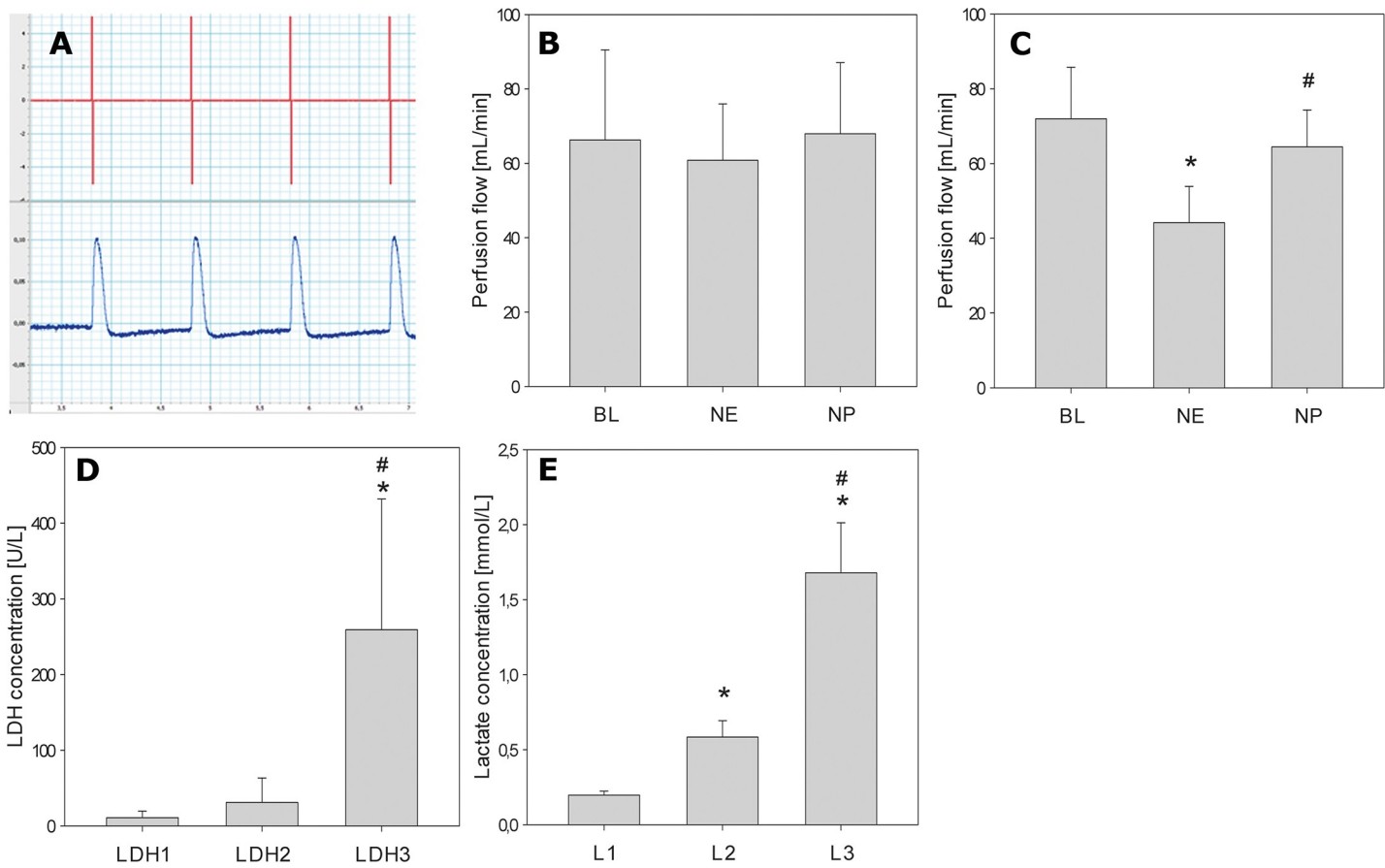

**Fig 2. Electrical stimulation, vitality tests, Lactate and LDH-measuring.** In **A**, a representative graph of the electrical stimulation (red) is shown and the muscular contraction follows promptly and regular (blue). The stimulation of the CAD was performed at the beginning and at the end of each experiment. **B** and **C** show alterations of the baseline perfusion flow (BL) after the administration of norepinephrine (NE) und sodium nitroprusside (NP) at the beginning of the experiment (**B**) and at the end of the experiment (**C**). Significant changes in the perfusion flow were detected when BL was compared to NE (*, p < 0.05) and NE to NP (#, p< 0.05). Data shown in **B-E** are mean and standard deviation. **D**: LDH was measured three times during the perfusion experiment (beginning, middle, end) and shows a significant increase from LDH3 to LDH2 (*, p < 0.05) and LDH3 to LDH1 (#, p< 0.05). **E**: Lactate concentration in the perfusion solution of the perfusion tub was measured at three different time points. Samples were taken at the beginning of the perfusion (L1), before exchange of the perfusion solution (L2) and at the end of the perfusion (L3). Data indicate a significant increase in the lactate concentration from L1 to L2, from L1 to L3 (*, p < 0.05) and also when L2 is compared to L3 (#, p < 0.05).

Indirect immunofluorescent analysis of the CAD sections revealed no nuclear staining pattern for HIF-1α. A weak signal of HIF was detected in the non-perfused CAD (Fig 3C) as well as in the perfused CAD (Fig 3D).

The representative staining of the hypoxia induction in equine cell cultures is shown in S1 Fig. Here, the indirect immunofluorescence analysis under hypoxic conditions clearly confirms the nuclear distribution pattern of HIF-1α.

## Discussion

This study for the first time presents a constant pressure perfused *ex vivo* model of the equine larynx. As different approaches for successful perfusions are available and all have advantages and disadvantages, we decided for a pressure constant and against a constant flow perfusion because the latter does not allow to test the autoregulatory mechanisms we examined.

The external electrical stimulation of the CAD in our study obtained an increase in the intramuscular pressure at the beginning and at the end of the perfusion. This pressure build-

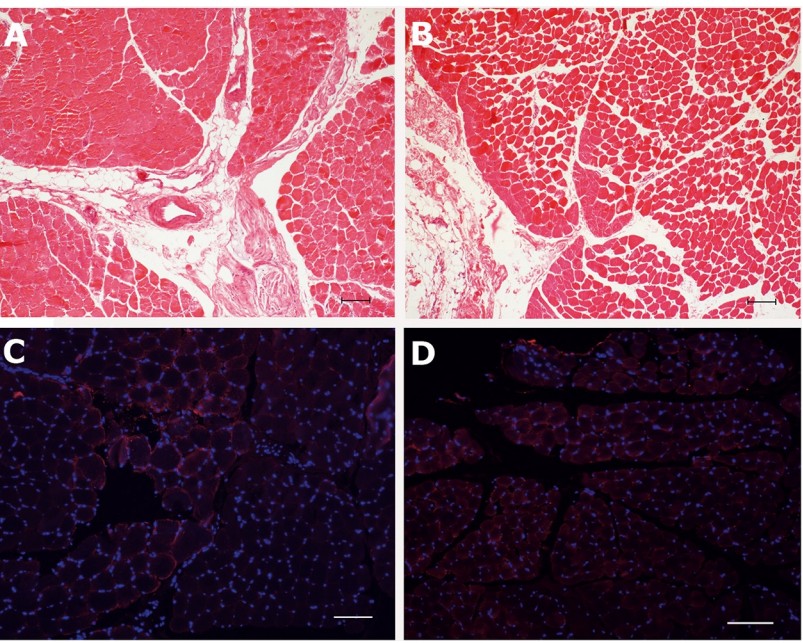

**Fig 3. Representative histology of the perfused CAD and evaluation of HIF-1α distribution. A** displays a Hematoxylin-Eosin staining of a non-perfused normal equine CAD-biopsy for histological comparison. In **B**, the perfused muscle tissue shows interstitial edema. Note the loosened connection of the muscle fibers with low amounts of poorly stained edema fluid. The perimysial borders are still distinguishable. **C** and **D** show immunofluorescent analysis of HIF-1α in the CAD-sections before and after perfusion. A diffuse weak staining within the muscle fibers can be observed (red). No intranuclear signal can be detected.

up indicated a contraction of the CAD and thereby the vitality of the CAD on each side, also visualized by an abduction of the vocal fold after 6 h perfusion duration. The test is simple and relatively easy to perform. Measuring intramuscular pressure provides information about the contractile force of a muscle [27, 28]. For testing the vitality of limb muscles after injuries, an external battery-powered electrostimulator was suggested [29]. However, the significance of external electrical stimulation should not be overinterpreted. Studies on human corpses showed that skeletal muscle excitations caused by mechanical stimulation can be observed for more than 13 hours post mortem [30]. For the assessment of the organs' vitality, additional testings aiming at the vascular level are needed.

NE is well known as a vasoconstrictive drug [31]. NP on the other hand acts as a vasodilating agent [32]. An explanation for the absent reaction to NE in one horse at the beginning of the perfusion might be the duration of the adaptation phase. A second larynx did not respond properly to the application of NP at the start of the perfusion. As there was a response to NE and NP in all larynges at the end of the perfusion, the ten-minute adaptation phase for the aforementioned two larynges might have been too short. In the literature, the duration of the adaptation phase for organ perfusion varies significantly from no adaptation phase to 60 minutes [12, 33, 34]. The reaction of the vessels to the application of NE and NP proves the full functionality of the vascular smooth muscle cells.

The measured values for lactate and LDH as accepted and widely used markers showed a statistically significant increase during the perfusion. Although increasing, all obtained values for LDH were well below maximum values in the references known from the literature [35]. Muñoz et al. [36] obtained maximum lactate values of 22.79 ± 6.75 mmol/L for horses after exercise. Even though, in the present study, lactate values at the end of the perfusion exceed

the reference range for horses, they distinctly remained below the aforementioned maximum values described in the literature. The first two measurements for LDH and lactate were in the reference range for horses. As the second sample for measuring was obtained approximately two hours after the beginning of the perfusion, it can be concluded that anaerobic metabolism took place after the second sample taking but no extensive cellular damage took place. For future studies, it is advisable to check lactate and LDH values more frequently in order to better evaluate the perfused organ.

It is known from the available case study using constant flow perfusion (8), that massive edema occurred during the perfusion. Macroscopically, the larynges perfused in the study presented here showed mild to moderate edema. This improved result was most likely achieved through the constant pressure perfused approach leading to less overperfusion. The interstitial edema might be discussed as a consequence because of the usage of a crystalloid perfusion solution. However, the tests carried out using NE and NP as well as the autoregulatory mechanisms prove the intact endothelial barrier and the functionality of the adjacent smooth muscle building the lamina muscularis.

To further substantiate our histological findings, HIF-1α-staining was performed. Although HIF-1α is an ubiquitously found protein, correct staining and critical interpretation of results are mandatory and a good positive control is a prerequisite for this to be sure not just to detect baseline values. This is why we aimed at establishing a hypoxia-induced control for the immunofluorescent experiments. To our knowledge, this is not described for equine cultures to date even though there are reports on mainly intracytoplasmic HIF-1α expression in other equine tissues [37, 38] and sarcoids as well as keratinocyte cultures [39].

Our $CoCl_2$-induced hypoxic cells displayed clear nuclear staining as described for other species [40]. The use of $CoCl_2$-induced cell cultures can therefore be recommended as a reliable positive control for HIF-1α also in equine culture systems. In this case, the stable HIF-1α-expression was achieved administering slightly higher concentrations of $CoCl_2$ than recommended [26]. Depending on the cell type used for the artificially induced hypoxia, it might be necessary to adjust the molarity of the $CoCl_2$-solution individually as cells from different origin can be differently susceptible to the treatment.

For the detection of HIF-1α in tissue sections, it is of note that HIF-1α possesses a short half-life period. One could possibly argue that the physiological reaction to hypoxic conditions, the following activation of HIF-1α and its initiating of counter regulatory processes (e.g. increased glycolysis, activation of VEGF) and the necessary adaptations took place before sample taking, thereby possibly missing the time frame in which HIF-1α was detectable.

However, we do conclude this is not the case because at least in a few samples or regions intranuclear detection of HIF-1α should have been apparent. We also monitored the surrounding connective tissue (e.g. perimysium) and this also lacked clear HIF-1α-staining restricted to the nuclei. Therefore, the well-controllable HIF-1α staining is a useful tool to check for a possible hypoxic condition which might be apparent.

The described *ex vivo*-approach towards a constant pressure perfused equine larynx physiologically mimics the *in vivo*-situation and different ways of further developing the model are conceivable. The advantages of the model are the tissue-conserving effects by preventing over- or underperfusion and the maintenance of the vessels´ autoregulation mechanisms instead of using a flow constant model. In order to further investigate RLN, additional studies are needed for e.g. testing abduction and adduction of the vocal folds. Also, this perfusion approach permits the investigation of known RLN-affected larynges. Cercone et al. showed the successful use of FES in vivo in horses with neurectomized recurrent laryngeal nerve [6]. With the model we present here, a controlled setup for different electrode configurations and their effect and optimized positioning would be beneficial for further investigating this technique.

Concerning the perfusion solution, we chose a relatively cost efficient and comparable setup. For testing longer perfusion time spans or more rigorous stimulation experiments, both the oxygen supply and to provide sufficient nutrients are core issues. For certain applications it might be necessary to be able to change from the continuous circulation system to a fresh supply with a drop tank to facilitate the flushing out of applied drugs. Nevertheless, autologous blood perfusion of isolated distal limbs in the horse is described and could hint at interesting further comparative studies in the field of larynx perfusion [41]. It might be of interest to translate the described model to a pulsed full blood perfusion system in order to enhance realistic conditions for the perfusion. Here, it has to be considered there are no ready to use blood products available for horses and using processed autologous blood comes with possible clotting or hemolysis.

Our model may serve as a solid base for the field of ex vivo larynx perfusion and there are numerous tests from other perfusion experiments in other organs that are worth to be considered in further studies.

## Supporting information

**S1 Fig. Positive control for HIF staining. A-H**: *In vitro* experiments to verify cross-reactivity of the anti-HIF-1α-antibody to equine cells. After incubating fibroblasts with 250 μM $COCl_2$ for 24 hours, HIF-1α translocation to the nucleus (**A**, in green) can be observed. The corresponding nuclear counterstain is displayed in **B** and the cytoplasmic marker vimentin (red) in **C** for enhanced visualization of the cells. All channels are merged in **D**. In **E-F**, the control (cells were not incubated with $COCl_2$ and stained simultaneously as described above) is shown. Here, the clearly weaker HIF-1α-staining (**E**) is not restricted to the nucleus but intracytoplasmically distributed. Appearance of nuclei and Vimentin staining resembles the induced cultures. In all images, nuclei are counterstained with Hoechst 33342 and all scale bars represent 100 μm. **A-H** are 20x magnifications in original.
(TIF)

## Acknowledgments

The authors thank J. Cheetham (Department of Clinical Sciences, Cornell University) for valuable comments on the project. At Leipzig University, we thankfully acknowledge the team of the equine clinic for their support, G. Lindner and G. Kölle for their help with preparing histological specimens and slides as well as measuring of laboratory parameters LDH and lactate and I. Kupfer for her help with the schematic drawing of the larynx.

## Author Contributions

**Conceptualization:** Sven Otto, Stefan Dhein, Christoph K. W. Mülling.

**Data curation:** Sven Otto, Jule K. Michler.

**Formal analysis:** Sven Otto.

**Funding acquisition:** Sven Otto, Christoph K. W. Mülling.

**Investigation:** Sven Otto, Jule K. Michler.

**Methodology:** Sven Otto, Jule K. Michler, Stefan Dhein.

**Project administration:** Stefan Dhein, Christoph K. W. Mülling.

**Resources:** Christoph K. W. Mülling.

**Software:** Christoph K. W. Mülling.

**Supervision:** Stefan Dhein, Christoph K. W. Mülling.

**Validation:** Sven Otto, Jule K. Michler.

**Visualization:** Sven Otto, Jule K. Michler.

**Writing – original draft:** Sven Otto, Jule K. Michler.

**Writing – review & editing:** Sven Otto, Jule K. Michler, Stefan Dhein, Christoph K. W. Mülling.

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
