## [Decision Letter · Decision Letter 0]

9 Apr 2021

PONE-D-21-04296

Development of a constant pressure perfused ex vivo model of the equine larynx

PLOS ONE

Dear Dr. Michler

Thank you for submitting your manuscript to PLOS ONE. After careful consideration, we feel that it has merit but does not fully meet PLOS ONE’s publication criteria as it currently stands. Therefore, we invite you to submit a revised version of the manuscript that addresses the points raised during the review process.

We look forward to receiving your revised manuscript.

Kind regards,

Maurizio Mandalà, Ph.D.

Academic Editor

PLOS ONE

Journal Requirements:

3)  Thank you for stating the following in the Competing Interests section:

[The authors have declared that no competing interests exist..

We note that you received funding from a commercial source: Med-El Company

**Comments to the Author**

1. Is the manuscript technically sound, and do the data support the conclusions?

Reviewer #1: Yes

Reviewer #2: Yes

2. Has the statistical analysis been performed appropriately and rigorously? 

Reviewer #1: Yes

Reviewer #2: Yes

3. Have the authors made all data underlying the findings in their manuscript fully available?

Reviewer #1: Yes

Reviewer #2: Yes

4. Is the manuscript presented in an intelligible fashion and written in standard English?

Reviewer #1: Yes

Reviewer #2: Yes

5. Review Comments to the Author

Reviewer #1: This manuscript presents for the readership of Plose One interesting results of an experimental study on tissue survival versus hypoxia of postmortem excised equine laryngeal organs during the course of a 6-hour in vitro perfusion. This manuscript represents original research. A pilot study on this project was published as a case study in the congress proceedings of a BMT (Biomedical Engineering) congress in 2013 (see reference 42).

The five larynxes were perfused with an oxygen-enriched, body-warm Tyrode saline solution at constant pressure via the superior laryngeal arteries. Venously, the irrigation solution ran into the laryngeal surrounding basin.Addition of vasoactive substances demonstrated vitality of blood vessels with preservation of vascular regulation of feeding arteries in both directions (vasoconstriction and vasodilatation) and their reversibility.

The hypoxia marker lactate dehydrogenase (LDH), which can be determined in the perfusion circuit, was determined three times (at the beginning, after changing the perfusion fluid (after approximately 2 h), and at the end of the experiment). The first two measurements were still in the upper normal range. At the end, LDH levels increased statistically significantly, but did not reach the level of a non-perfused larynx. Thus, anaerobic metabolism occurred after the 2h value. As the authors correctly note, continuous LDH measurements would have been helpful to more accurately document the transition to anaerobic metabolism.

In addition, tissue hypoxia was determined at the end of the experiment in a very elaborate manner using histological examinations of the only vocal fold abductor muscle of the equine larynx, the cricoarytenoideus dorsalis muscle (CAD), with the aid of an immunofluorescent analysis of hypoxia-inducible factor (HIF-1α) adapted specifically for this study.For this purpose, an additional positive control had to be established in the form of a cell culture. As a result, severe hypoxia damage of the muscle fibers after 6 h of perfusion could be excluded. This information is of great importance for the future use of such an in vitro model as a substitute for in vivo experiments, e.g. for electrical stimulation of vocal fold motility.

A major future application of such an in vitro perfusion model of mammalian larynxes will be its use for surgery simulation and for testing new dynamic therapeutic approaches, such as functional electrical stimulation (FES). This experimental study demonstrated that electrical stimulation of still-vital nerve endings in CAD muscle at the end of the experiment could still elicit a muscle contraction comparable to the initial one, as measured by an intramuscular pressure sensor. In my eyes, this is the most significant statement of this work. It is therefore recommended that the authors emphasize this aspect more strongly in the discussion. The stimulation parameters used cannot trigger direct muscle fiber stimulation, thus after 6 h the nerve-muscle unit was still vital. For the future experiments mentioned by the authors to measure the stimulation current propagation in CAD depending on the location, configuration, and size of possible stimulation electrodes, it is of importance how long such a model remains vital under recurrent or continuous electrical stimulation. In my understanding, the supply of both oxygen and substrate nutrients to the tissue then plays a role. This should be added when discussing the change to a fresh blood perfusion model.In addition to current intensity, pulse rate, and pulse duration, the authors were asked to specify the time for which electrical stimulation was applied at the beginning and end of each experiment. For future experiments, it would be important from the point of view of future clinical questions to investigate tissue hypoxia as a function of stimulation duration and current input and to continue the experiment in each case until the model can no longer be used. From this, the maximum useful life of such a model could be determined.

Overall, however, this does not diminish the importance of this work, which is a milestone of basic research towards a replacement model for in vivo animal experiments on laryngeal electrostimulation.

All Experiments, statistics, and other analyses are performed to a high technical standard and are described in sufficient detail.The conclusions, except for the additions to electrostimulation listed above, are adequately presented and supported by the data. The article is presented in an intelligible fashion and is written in standard English. The research meets all applicable standards for the ethics of experimentation and research integrity. The article adheres to the appropriate reporting guidelines and community standards for data availability and can be recommended for publication in Plos One with minor revisions.

Reviewer #2: Interesting paper. It appears to be a well designed setup which will hopefully allow further investigation into RLN. Just a few comments.

Page 5 line 58, Page 6 line 74- Please elaborate on your experimental setup. I am quite familiar with developing an ex vivo perfusion apparatus and was having a hard time figuring out exactly how the perfusion system works with your paper.

What arteries were cannulated? Passive drainage (allowing the veins to drain to the tub below)? Was the larynx immersed in the solution? please be more specific

Page 6 line 85- “A pressure transducer (Transducer Model SPR-524, Millar Instruments, Inc., Houston, Texas, USA) was inserted into the medial part of the CAD... The electrical impulses and the change of the intramuscular pressure during contraction were recorded by a PowerLab”

Unfortunately a twitch or local contraction as noted on the pressure transducer does not always result in movement of the laryngeal musculature.

Was the movement of the CAD noted visually or just via the transducer?

6. PLOS authors have the option to publish the peer review history of their article (what does this mean?). If published, this will include your full peer review and any attached files.

Reviewer #1: **Yes: **Prof. Andreas H. Mueller, M.D.

Reviewer #2: No

---

## [Author Response · Author response to Decision Letter 0]

23 Apr 2021

Comments raised by Reviewer 1 (Prof. Müller)

First, we would like to explicitely thank you for your detailed and appreciative review of our manuscript.

1. Rev. 1 asked for a revision of the discussion towards the CAD contraction at the beginning/end of the experiment. We rewrote this by adding the abducted vocal fold and the perfusion duration (lines 294-295). This is indeed a central finding of this study, but as it already has a front row seat in the discussion, we don´t want to overinterpret this as it is the first study to report such data in a modest number of ex vivo larynges and we tend to judge our data as critical as possible.

2. The next point Rev. 1 recommends is to add information to the outlook about fresh blood perfusion in hindsight of oxygen supply and substrate nutrients. We were happy to do this and rearranged the text accordingly (lines 364-373).

3. Rev. 1 then asked to specify the duration of the electrical stimulation at the beginning and at the end of the experiment which was 8 s (minimum). We supplemented the manuscript with this (material and methods section, line 138).

We would like take the opportunity to comment on this factor of central importance to you: of course, one of the logical next steps would consist of stress testing the model by moving the vocal fold for a certain period of time/try different pulses etc. to determine the point where this model reaches its capacity. This will require quite a number of organs, an intelligent sub-grouping and will be complicated by the overall lacking data in this field for ex-vivo larynx perfusion towards the strength of the muscular answer, what is considered a weak answer (etcpp), where the cut-off is... Nevertheless, we absolutely agree that this is something highly desirable!

We intendedly used this test to prove our approach as resulting in a functional model and learned during our experiments what a hallmark this relatively simple test presents.

Comments raised by Reviewer 2

1. Rev.2 asked for a more detailed description of our set-up. We extended the caption of Fig 1 that explains the experimental set-up and inserted an additional schematic drawing of an equine larynx in order to better illustrate the catheterization.

The corresponding caption reads now (lines 120-134):

A: The perfusion system (technical drawing) consists of a closed circuit. Red color indicates the hot water circuit, blue represents the perfusion circuit, gassed with carbogen. For active perfusion, the arteries were catheterized, passive venous outflow drains into the perfusion tub. The gassing tubes (one in the supply tank 1 and one in the perfusion tub) are denoted with an asterisk. The perfusion flow was measured using a flowmeter (F). In the inset photograph, a dorsal view of the perfusion setup in situ can be observed. The larynx, immersed in perfusion solution, is placed on a larynx retainer to stabilize its position in the perfusion tub. Both cranial thyroid arteries were catheterized (intravenous catheter, 18 gauge). For pH-control, a pH-meter is added (marked with an arrow).

B: Schematic drawing of an equine larynx, lateral aspect. Relevant structures are the cricoarytenoideus dorsalis muscle [CAD, a)], the common carotid artery [b) running parallel to the external jugular vein], the catheterized arterial branch [c), cranial thyroid artery] followed by its ramification into the ascending pharyngeal artery (d) and the laryngeal branch (e). The latter two vessels nourish the CAD. f) Recurrent laryngeal nerve reaches the larynx as caudal laryngeal nerve.

Thank you very much for this helpful comment, we hope to have addressed it sufficiently.

2. The next comment by Rev. 2 addresses the electrical stimulation of the CAD. The contraction of the CAD can be seen by an abduction of the vocal fold. This was judged visually. We did not develop a comparable measurement approach for this and therefore it is stated descriptively in the text. To improve this passage, we refined the results accordingly (lines 225-226).

---

## [Editor Report · Decision Letter 1]

28 Apr 2021

Development of a constant pressure perfused ex vivo model of the equine larynx

PONE-D-21-04296R1

Dear Dr. Michler,

We’re pleased to inform you that your manuscript has been judged scientifically suitable for publication and will be formally accepted for publication once it meets all outstanding technical requirements.

Kind regards,

Maurizio Mandalà, Ph.D.

Academic Editor

PLOS ONE
---

## [Editor Report · Acceptance letter]

7 May 2021

PONE-D-21-04296R1 

Development of a constant pressure perfused *ex vivo* model of the equine larynx 

Dear Dr. Michler:

I'm pleased to inform you that your manuscript has been deemed suitable for publication in PLOS ONE. Congratulations! Your manuscript is now with our production department. 

Kind regards, 

on behalf of

Dr. Maurizio Mandalà 

Academic Editor

PLOS ONE